# 3C-GAN: An condition-context-composite generative adversarial networks for generating images separately

## Abstract

We present 3C-GAN: a novel multiple generators structures, that contains one conditional generator that generates a semantic part of an image conditional on its input label, and one context generator generates the rest of an image. Compared to original GAN model, this model has multiple generators and gives control over what its generators should generate. Unlike previous multi-generator models use a subsequent generation process, that one layer is generated given the previous layer, our model uses a process of generating different part of the images together. This way the model contains fewer parameters and the generation speed is faster. Specifically, the model leverages the label information to separate the object from the image correctly. Since the model conditional on the label information does not restrict to generate other parts of an image, we proposed a cost function that encourages the model to generate only the succinct part of an image in terms of label discrimination. We also found an exclusive prior on the mask of the model help separate the object. The experiments on MNIST, SVHN, and CelebA datasets show 3C-GAN can generate different objects with different generators simultaneously, according to the labels given to each generator.

## 1 Introduction

Recently Generative adversarial network (GAN) is gaining much attention because it has shown promising results for generating natural images (Goodfellow et al. (2014); Goodfellow (2016)). Still, it is limited in generating globally coherent, or high-resolution images. Therefore, to have GAN generate better quality images is the main research topic. There have been many works tackling this topic, including using novel network structures, (Radford et al. (2015)), Denton et al. (2015)), using novel objective functions for training ( Zhao et al. (2016), Arjovsky et al. (2017), Gulrajani et al. (2017)), using multi-stage network generation (Im et al. (2016), Kwak & Zhang (2016), Yang et al. (2017)), applying 3D structure information for image generation( Wang & Gupta (2016)).

Another main research avenue of GAN is to make a generator more controllable, that is, the generator can generate the image given the criterion we want. Conditional GAN serves this purpose by ensuring the generation is conditional on the given criterion (Odena et al. (2016), Mansimov et al. (2015), Reed et al. (2016a), Kaneko et al. (2017), Zhu et al. (2016)). Specifically, in these conditional models, the generator not only accepts the noise code, but also a conditional code, either from a one-hot representation of class labels, or a sentence embedding from a text, or an image, such that this given side information encodes the criterion, and the generated images are conditional on that. Aside from the conditional model, that is a supervised approach, there are models employed an unsupervised approach. InfoGAN (Chen et al. (2016)) does not need any labels: it discovered the high-level meaning of the image purely from data, and assign the meanings to its noise code. For example, for MNIST dataset, InfoGAN model learn to have one code control the width of the generated digits, and one code controls the rotation of the digits. As an unsupervised approach, InfoGAN discovered the criteria that we might think is useful for the dataset.

Both the supervised and unsupervised models are one-generator approach, there is another multi-generators approach that also aims to make the model more controllable (Kwak & Zhang (2016) Yang et al. (2017)). They generate images part-by-part, such that each part has its semantic mean-

ing, and combing all parts results in a realistic image. Since a part of an image has some persistent characteristics, these model learn which parts of the data should be generated together in an unsupervised way. In addition, the model of Yang et al. (2017) includes a transformer network that learns an affine transformation for the foreground in an image. Since each part of an image is generated by different generators, these model provide a high-level meaning of the generation for each generator and possess a finer control over original GAN.

We develop a model that also aims to improve the controllability over original GAN. Specifically, our model is a multi-generators approach, where one generator is a "context" generator that does not take any conditional labels as input, and another is a conditional generator that takes its label information. The conditional generator generates the part of the image that is related to its labels, whereas the context generator generates the other part of the image. Both generators learn where to generate (a mask), and what to generate (an appearance). The context generator shared its input codes with the conditional generator so it "knows" what context it is on when generating its own part. However, there is no restriction for the conditional generator to not generate the context. We propose a cost function on the label discrimination to penalize the conditional generator to do so. In addition, we proposed an exclusive prior on the mask such that any pixels in an image should be generated from one generator only. All in all, our contributions can be listed as follows.

1. We are the first one showing that by applying label information, the model and the cost function we proposed, we can have a generator to learn to generate the part of the image only related to the label.

2. We show that a simpler multi-generators structure, without using Recurrent Neural Net to generate different part of the image subsequently, works to generate images part-by-part simultaneously.

## 2 METHOD

The algorithm we proposed is based on Wasserstein GAN, conditional GAN, and model an image with a layered structure, we introduce them in the background section.

### 2.1 BACKGROUND

#### 2.1.1 WASSERSTEIN GAN

A generative adversarial network (GAN) consists of two neural networks trained simultaneously and in opposition to one another. Assume a real sample $\mathbf{x}$ follows an unknown distribution $p_{data}$. The generator network G takes as input a random code vector $\mathbf{z}$ and output the fake data $\mathbf{x}_f$, such that $\mathbf{x}_f = G(\mathbf{z})$, whereas the discriminator network D takes as input either a training sample or a synthesized sample from G and outputs a probability that the input is real or fake. The discriminator is trained to maximize the probability of assigning the correct source to both training samples and samples from G, where the generator is trained to minimize the probability that D assigns its generated sample as fake. The objective function is

$$\min_{\theta_G} \max_{\theta_D} \left( E_{\mathbf{x}_r \sim p_{data}(\mathbf{x}_r)}[\log D(\mathbf{x}; \theta_D)] + E_{\mathbf{z} \sim p_{\mathbf{z}}(\mathbf{z})}[\log(1 - D(G(\mathbf{z}; \theta_G); \theta_D))] \right) \quad (1)$$

, where $D$ and $G$ are parameterized by $\theta_D$ and $\theta_G$ respectively.

Compared to original GAN, Wasserstein GAN is more stable and resilient to hyper-parameters changes (Arjovsky et al. (2017)). Another advantage over original GAN is the cost function of it decreases steadily among training. Wasserstein GAN also has two networks D and G that are trained simultaneously and adversarially. The main difference is instead outputting a probability, D outputs a real value. The objective function becomes:

$$\min_{\theta_G} \max_{\theta_D} \left( E_{\mathbf{x}_r \sim p_{data}(\mathbf{x}_r)}[D(\mathbf{x}_r; \theta_D)] - E_{\mathbf{z} \sim p_{\mathbf{z}}(\mathbf{z})}[D(G(\mathbf{z}; \theta_G); \theta_D)] \right) \quad (2)$$

The difference between the first and the second term is the estimated Wasserstein distance ($W$). While D maximizes the distance, $G$ minimizes it. Importantly, $D$ needs to be a 1-Lipschitz function so that the estimation is accurate. A penalty term involves the first-order derivative of $D$ is added to the objective function to encourage this requirement. Readers can find the details of it in Gulrajani et al. (2017).

### 2.1.2 CONDITIONAL GAN

Conditional GAN is an extension of GAN that makes it conditional on the prior information of the data, and here we focus on a type of the conditional GAN that is conditional on image labels or attributes. More specifically, we applied the conditional GAN model proposed in Odena et al. (2016) (AC-GAN). Here we briefly introduce this model. Assume a real sample $x$ and its label $\mathbf{l}_x$ follows an unknown distribution $P_{data}$. The label is represented as a one-hot representation. The generator $G$ takes as input a random code vector $\mathbf{z}_u$ concatenated with a one-hot representation of a random label $\mathbf{z}_l$ to generate a fake sample, such that $\mathbf{z}_c = [\mathbf{z}_u, \mathbf{z}_l], \mathbf{x}_f = G(\mathbf{z}_c)$. To encourage $G$ to generate a sample that is conditional on $\mathbf{z}_l$, we have an auxiliary classifier $C$, such that $D$ and $C$ share all layers except the last layer. While $D$ outputs the Wasserstein distance mentioned above, $C$ takes the input sample and its label, and outputs a cross-entropy loss between the label and the probability distribution over the class labels. The objective function becomes:

$$\min_{\theta_G} \max_{\theta_D} \min_{\theta_C} \left( E_{\mathbf{x} \sim p_{data}(\mathbf{x})}[D(\mathbf{x}; \theta_D)] - E_{\mathbf{z}_c \sim p_{\mathbf{z}_c}(\mathbf{z}_c)}[D(G(\mathbf{z}_c; \theta_G); \theta_D)] \right) +$$
$$\left( E_{\mathbf{x}, \mathbf{l}_x \sim p_{data}(\mathbf{x}, \mathbf{l}_x)}[-\log C(\mathbf{l}_x | \mathbf{x}; \theta_C)] + E_{\mathbf{z}_u, \mathbf{z}_l \sim p_{\mathbf{z}_c}(\mathbf{z}_u, \mathbf{z}_l)}[-\log C(\mathbf{z}_l | G(\mathbf{z}_c; \theta_G); \theta_C)] \right)$$
$$(3)$$

Where the first two terms correspond to the Wasserstein distance in equation 2, the third term corresponds to the cross-entropy loss for a real image ($L_c^r$), and the fourth term corresponds to the cross-entropy loss for the fake image ($L_c^f$).

### 2.1.3 LAYERED-STRUCTURE MODELING FOR IMAGES

Since an image is taken from our 3D world, it usually contains a layered structure. Modeling an image as with a mask is common. An example of that is a two-layered foreground/background modeling,

$$\mathbf{x} = \mathbf{f} \odot \mathbf{m} + \mathbf{b} \odot (1 - \mathbf{m})$$

, where $\mathbf{f}$ is the foreground image, $\mathbf{b}$ is the background, $\mathbf{m}$ is the mask for the foreground, and $\odot$ is element-wise multiplication. This modeling has already been applied in synthesize natural images (Isola & Liu (2013), Reed et al. (2016b), Yan et al. (2016)). This modeling has also been applied to videos (Darrell & Pentland (1991), Wang & Adelson (1994), Jojic & Frey (2001), Kannan et al. (2005), Vondrick et al. (2016).

Recently there are two GAN models also applied the foreground/background layer setting Kwak & Zhang (2016), Yang et al. (2017), the generation process for them is the second layer (foreground) is generated subsequently after the first layer (background). In contrast, we model an image as a context layer and a conditional layer and generate the two layers together.

## 2.2 CONDTION-CONTEXT-COMPOSITE GAN (3C-GAN)

### 2.2.1 MODEL DEFINITION

Our model is based on the conditional GAN and Wasserstein GAN. It has one Discriminator $D$, one auxiliary classifier $C$, but two generators $G_1$ and $G_2$. $G_1$ is an ordinary generator, but $G_2$ is a conditional generator. While $G_1$ generates the context of an image, $G_2$ generates the part of an image that is conditional on its input label. The final output, $(X_f)$, is the composite image from $G_1$ and $G_2$. The model definition is shown below.

$$\mathbf{m}_1, \mathbf{f}_1 = G_1(\mathbf{z}_u, \mathbf{z}_l)$$
$$\mathbf{m}_2, \mathbf{f}_2 = G_2(\mathbf{z}_u, \mathbf{z}_v, \mathbf{z}_l)$$
$$\mathbf{m}_1^n, \mathbf{m}_2^n = \text{softmax}(\mathbf{m}_1, \mathbf{m}_2)$$
$$\mathbf{o}_1 = \mathbf{f}_1 \odot \mathbf{m}_1^n$$
$$\mathbf{o}_2 = \mathbf{f}_2 \odot \mathbf{m}_2^n$$
$$\mathbf{x}_f = G_c(\mathbf{z}_u, \mathbf{z}_v, \mathbf{z}_l) = \mathbf{o}_1 + \mathbf{o}_2$$

Figure 1 shows the architecture of 3C-GAN. A part of input noise codes ($\mathbf{z}_u$) that is for context generation. $\mathbf{z}_u$ is shared for both $G_1$ and $G_2$ because $G_2$ also needs to know the context to generate.

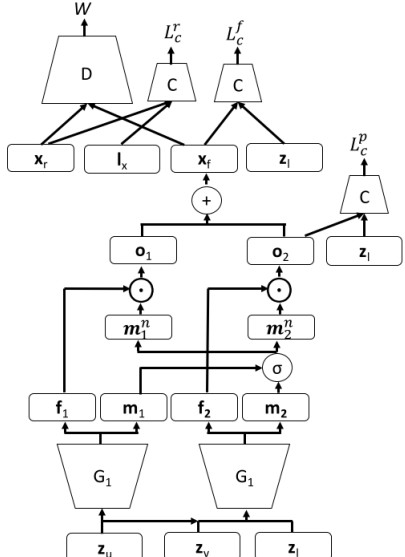

Figure 1: The architecture of 3C-GAN. The $\sigma$ stands for softmax normalization, and the $\odot$ stands for element-wise multiplication. Note that classifier network $C$ is reused three times for different input and label pairs.

The same $\mathbf{z}_u$ for $G_1$ and $G_2$ inputs enable them to "communicate" with each other. At the same time, $G_2$ takes additional input codes ($\mathbf{z}_v$) and label codes ($\mathbf{z}_l$). $\mathbf{z}_v$ let $G_2$ has additional power to control over the conditional part.

We denote $G_c$ as the composite model of $G_1$ and $G_2$. Each generator generates a mask ($\mathbf{m}_1, \mathbf{m}_2$) and an appearance ($\mathbf{f}_1, \mathbf{f}_2$). The masks generated from $G_1$ and $G_2$ are further normalized by softmax. Each appearance element-wisely multiplies by each mask and summed together. This way the mask controls to generate which part of an image. The composite fake image $x_f$ is then given to discriminator and the optimization is as same as equation 3.

There is an auxiliary classifier $C$ helps the model conditional on its label. As AC-GAN, Both $\mathbf{x}_r$ and its label $\mathbf{l}_x$ pair, and $\mathbf{x}_f$, $\mathbf{z}_l$ pair are given to $C$ to compute the classification loss. The losses for two pairs are the ($L_c^r$) and ($L_c^f$) in equation 3, respectively. The loess encourage $G_c$ as a whole to be conditional on the label; However, they do not encourage $G_2$ to be a conditional generator. Intuitively, if $G_2$ generates the conditional part, $G_2$ should have to be discriminated as the same label as $G_c$; therefore, we proposed an additional classification loss ($L_c^p$) for the input pair $\mathbf{o}_2$, $\mathbf{z}_l$, given the same classifier $C$. That is:

$$L_c^p = E_{\mathbf{z}_u,\mathbf{z}_v,\mathbf{z}_l}\left[-\log C(\mathbf{z}_l|G_2(\mathbf{z}_u,\mathbf{z}_v,\mathbf{z}_l))\right] \tag{4}$$

In practice, there is no restriction on the structure of $D$ and $G$. To have good generating quality and fast convergence, we applied the structure of DCGAN (Radford et al. (2015)) . To save computational power, $G_1$ and $G_2$ can share all the layers except the first(bottom) layer, that is a fully-connected layer computing the 4*4 feature map from the input code.

Besides the basic GAN cost and conditional cost (classifier loss), there are two additional costs that are essential for our purpose.

### 2.2.2   LABEL DIFFERENCE COST

In the model defined above, $G_2$ is not restricted to generate the unconditional part of an image, that is, $G_2$ could take the job $G_1$ does. In the extreme case, $G_2$ could generate all images, that is, $m_1^n$ is zero everywhere. Here we proposed a cost to penalize $G_2$ such that it generates only the succinct part when changing the condition of an image.

$$L_{ld} = E_{\mathbf{z}_u,\mathbf{z}_v,\mathbf{z}_l}\left[\sum |G_c(\mathbf{z}_u,\mathbf{z}_v,\mathbf{z}_l) - G_c(\mathbf{z}_u,\mathbf{z}_v,\mathbf{z}_l^f)|\right] \tag{5}$$

, where $\mathbf{z}_l^f$ (label flip) is any label code that is different from $\mathbf{z}_l$, and the summation is over all pixels in an image. Since only $G_2$ is accessible to the condition information, $G_2$ is forced to generate the succinct part of condition changes when we apply this cost.

### 2.2.3 EXCLUSIVE PRIOR

In the above model, except that $\mathbf{m}_1 + \mathbf{m}_2 = 1$, we do not have any prior on these masks. In practice, if one pixel is generated by a generator, we want it to take a full responsibility for generating it, that is, the value of its mask for that pixel should be close to 1. Here we proposed an exclusive prior on the two masks.

$$L_{ex} = E_{\mathbf{z}_u, \mathbf{z}_v, \mathbf{z}_l} \left[ \sum |\mathbf{m}_1 \odot \mathbf{m}_2| \right] \tag{6}$$

, where the summation is over all pixels in an image. We found adding this prior makes $G_1$ and $G_2$ generate a part of an image separately.

### 2.2.4 OBJECTIVE FUNCTION

The overall loss function for 3C-GAN is:

$$L_{all} = W + L_c^r + L_c^f + L_c^p + \alpha L_{ld} + \beta L_{ex} \tag{7}$$

The definition of $W, L_c^r, L_c^f$ is the same as that in conditional GAN. $L_c^p$ is defined in the model definition for 3C-GAN. $\alpha$ and $\beta$ are additional hyper-parameters that need to be tuned. In practice, we first tune $\alpha$ with $\beta$ fixed to 0, find the best $alpha$, then tune $\beta$ with the best $\alpha$. We can tune both $\alpha, \beta$ qualitatively or quantitatively. If we visualize $x_f$, we can see if it is generated according to labels. If not, $alpha$ for that setting is too high. Also, if we visualize $m_1$ or $m_2$, we can see if they are near zero or one. On the other hand, if the loss $L_c^f$ and $L_c^p$ becomes too high, we know the $alpha$ is set too high.

## 3 EXPERIMENT

We conduct experiments on three datasets 1)MNIST LeCun et al. (1998) 2)SVHN Netzer et al. (2011) 3)CelebA Liu et al. (2015). To display our method can separately generate the part of an image with its label, we modify MNIST data making the image background a uniform grayscale value between [0, 200], and resize it to 32*32. We name this modified version "MNIST-BACKGROUND". For SVHN, we use the extra set of the cropped digits. For CelebA, we focus on the label of smile/not smile, and use the set of aligning face. We implement our method based on open source code[1]. The details of parameters of each experiment will be discussed in each section.

### 3.1 RESULTS FOR MNIST-BACKGROUND DATASET

MNIST-BACKGROUND is similar to MNIST, with 10 different digits and 50000 images for training. We set the channel number of the model to be 32, and the input noise code number to be 30. There are 15 context codes ($\mathbf{z}_u$), 15 object codes ($\mathbf{z}_v$), and 10 label codes ($\mathbf{z}_l$). Therefore, the input dimensionality for $G_1$ is 15 and for $G_2$ is 40. The weight for label difference cost ($L_l d$) is set to be 1 ($\alpha = 1$), and the weight for exclusive cost ($L_e x$) is set to be 0 ($\beta = 0$). The two generators shared all structure except the bottom layer, and model is trained for 100000 iterations.

The results are shown in figure 2 to figure 4. The three figures are generated with the same input noise code so we can compare them. The samples are generated conditional on digit class 0, 1, ... to 9. There are totally 128 samples, each digit sample repeats 12 times, and the rest is fed with digit class 0. Since this dataset is simple, the training is converged, and the model shows good generation result (2). In addition, we show the results generated by $G_1$ and $G_2$ in figure 3 and figure 4. For visualization purpose, we let a checkerboard pattern $\mathbf{c}$ composite with the output from $G1$, such that figure 3 shows $\mathbf{f}_1 \odot \mathbf{m}_1^n + \mathbf{c} \odot (1 - \mathbf{m}_1^n)$. Similarly, figure 4 shows $\mathbf{f}_2 \odot \mathbf{m}_2^n + \mathbf{c} \odot (1 - \mathbf{m}_2^n)$. The obvious checkerboard pattern in both figures means the mask is zero in those pixels. We can see $G_1$ only generate the context while $G_2$ generate the part of the image related to the label.

---

[1]https://github.com/igul222/improved_wgan_training

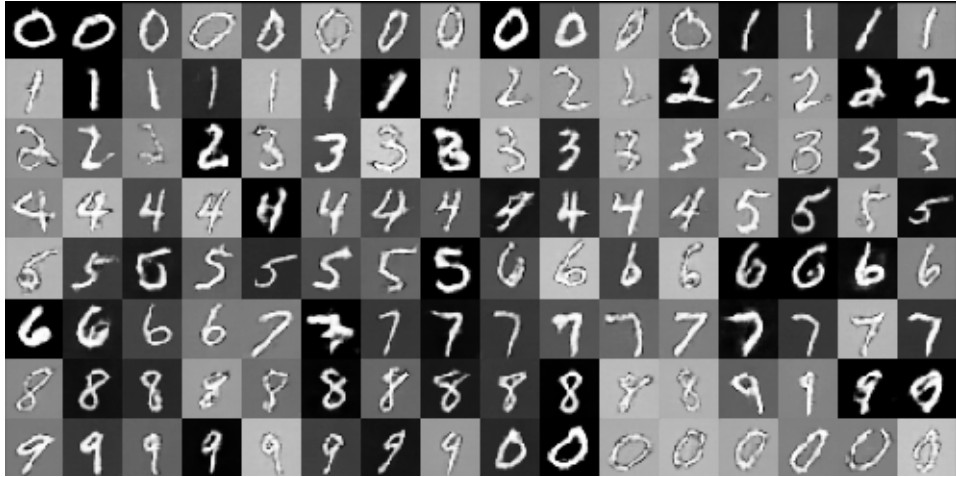

Figure 2: composite generated samples $x_f$ for MNIST-BACKGROUND dataset

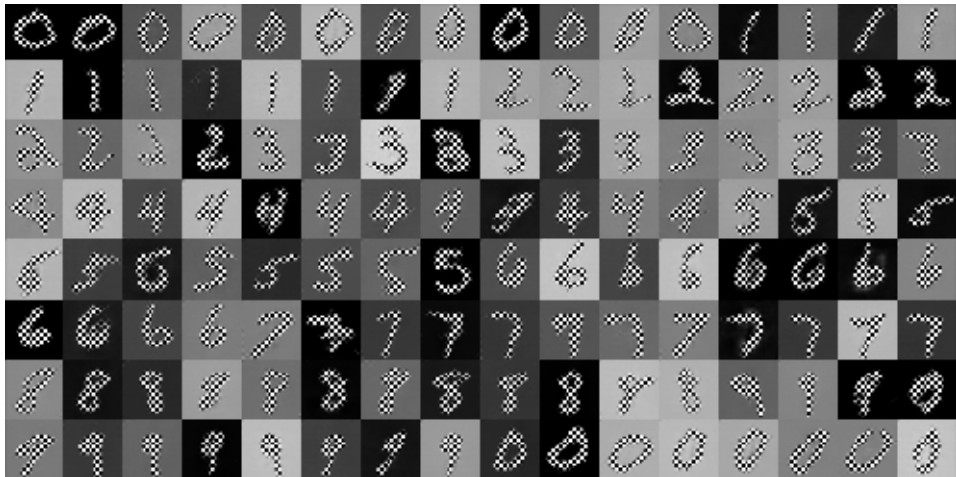

Figure 3: generated samples from $G_1$ for MNIST-BACKGROUND dataset

## 3.2 RESULTS FOR SVHN DATASET

SVHN is also a dataset with 10 different digits classes. In an image, a digit with the class label is in the center; however, there could be digits on the side of an image that does not related to the image's label. When training on this dataset, we set the channel number of the model to be 64, and the input noise code number to be 64. There are 32 context codes ($\mathbf{z}_u$), 32 object codes ($\mathbf{z}_v$), and 10 label codes ($\mathbf{z}_l$). Therefore, the input dimensionality for $G_1$ is 32 and for $G_2$ is 74. $\alpha$ is set to 10, and $\beta$ is set to 0.5. The two generators also shared all structure except the bottom layer, and model is trained for 100000 iterations.

The results are shown in figure 5 to figure 7. The setting is the same as before: the three figures are generated with the same input noise code, and the samples are generated conditional on digit class 0, 1, ... to 9. Figure 5 shows the model generates fairly reasonable images. In figure 6 and figure 7, we see $G_2$ only generates the center digit in an image that is labeled, other digits are left to $G_1$. Compared to the results for MNIST-BACKGROUND, one major difference is the digits in $G_2$ include the context other than the digit itself. We suspect this happens because sometimes the digits are dark while the background is light, and sometimes vice versa. And the digits generated by $G_2$ need to include the context around to have the classifier $C$ give it a high probability.

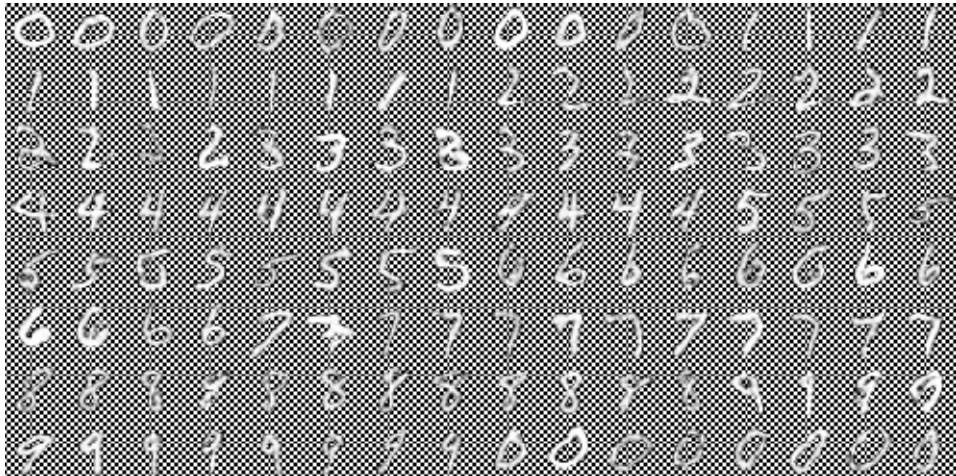

Figure 4: generated sample from $G_2$ for MNIST-BACKGROUND dataset

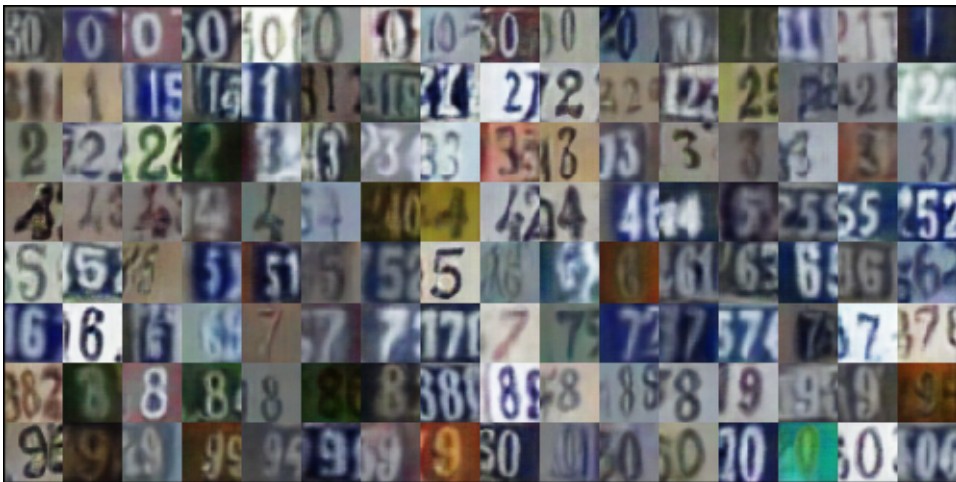

Figure 5: composite generated samples $x_f$ for SVHN dataset

### 3.3 RESULTS FOR CELEBA DATASET

There are 20599 images on this dataset. We pre-process the dataset by resizing the images to 64-by-64. We choose the attribute of smiling as image label for the conditional generator in our model because the number of images of smiling and not-smiling is similar. When training on this dataset, we set the channel number of the model to be 128, and the input noise code number to be 128 ( 64 context codes ($\mathbf{z}_u$), 64 object codes ($\mathbf{z}_v$), and 2 label codes ($\mathbf{z}_l$). $\alpha$ is set to 100, and $\beta$ is set to 0.5.

The results are shown in figure 8 to figure 10. There are 100 samples, with the first fifty samples belong to the class "not smiling", and the second fifty belongs to "smiling". The results show $G_2$ focus on face generation while $G_1$ focus on background generation. Since the condition of smiling/not smiling is determined by the face, it makes sense that $G_2$ take the responsibility to generate the faces. However, the separation is not as clear as the results on SVHN and MNISTB. We suspect this happens because we use the attribute smiling/not smiling as a condition, which only affects a small fraction of pixels in an image, and that is not strong enough to provide $G_2$ a solution to generate the whole face.

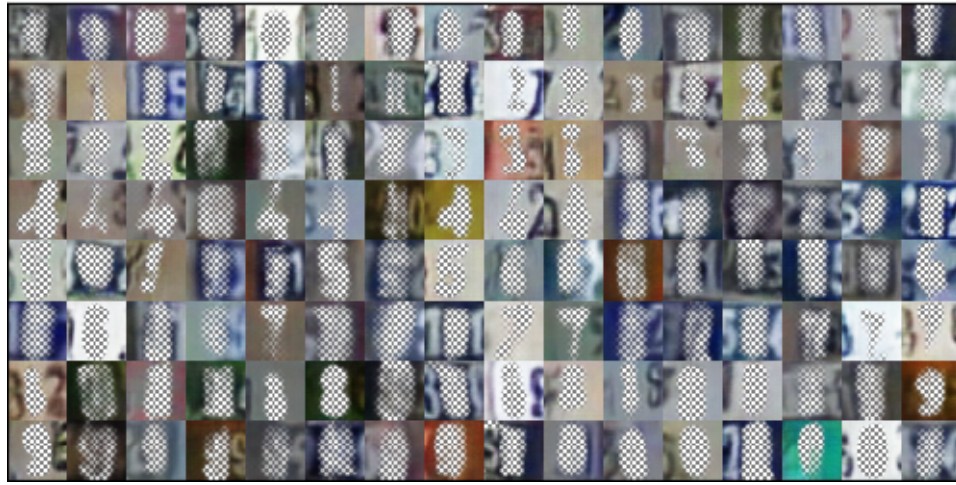

Figure 6: generated samples from $G_1$ for SVHN dataset

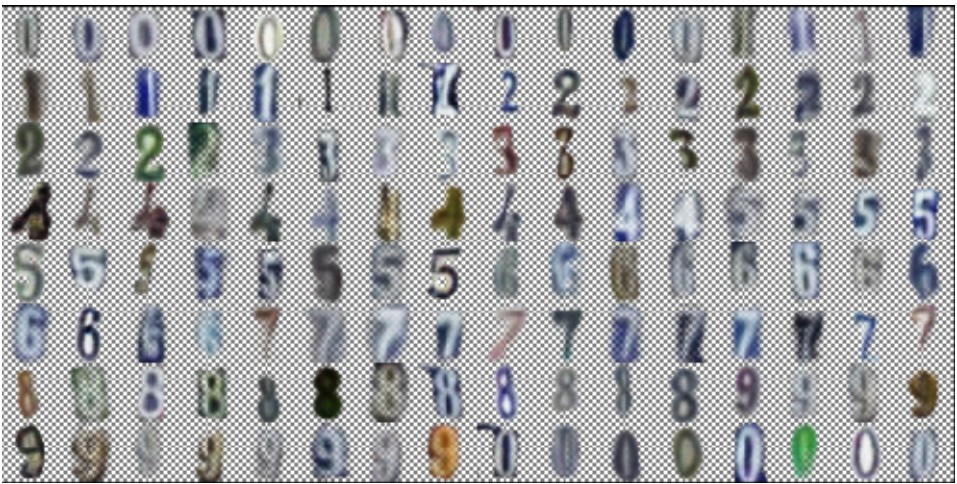

Figure 7: generated sample from $G_2$ for SVHN dataset

## 4 CONCLUSION

In this paper, we present a novel GAN structure that has one generator generates the context while the other conditional generator generate the part of the image based on the label it has. Compared to previous multi-generators model, our model has fewer parameters and generate the different part simultaneously. In addition, we proposed a new cost that makes the conditional generator learns to generate only the essential part of the condition changing. Also, we proposed an exclusive prior so that the two generators do not generate the same pixel. Experiments show our model separated the data as mentioned, and therefore; provide more controllability over original GAN.

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

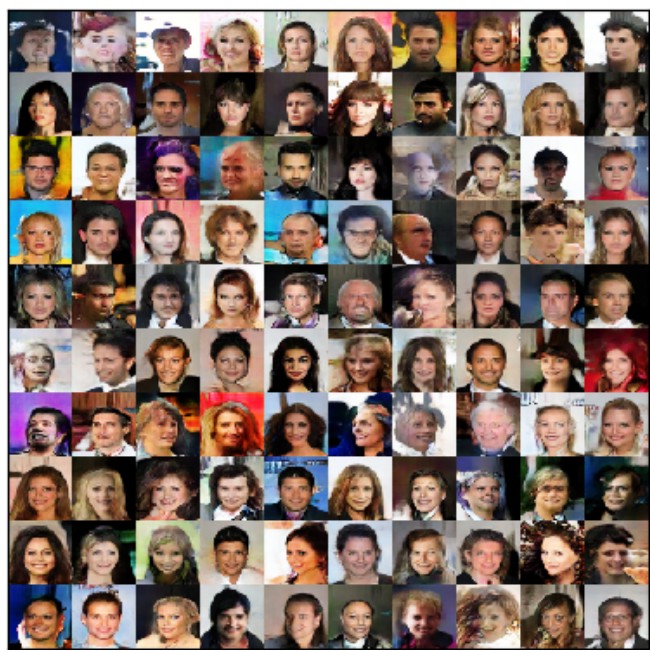

Figure 8: composite generated samples $x_f$ for CelebA dataset

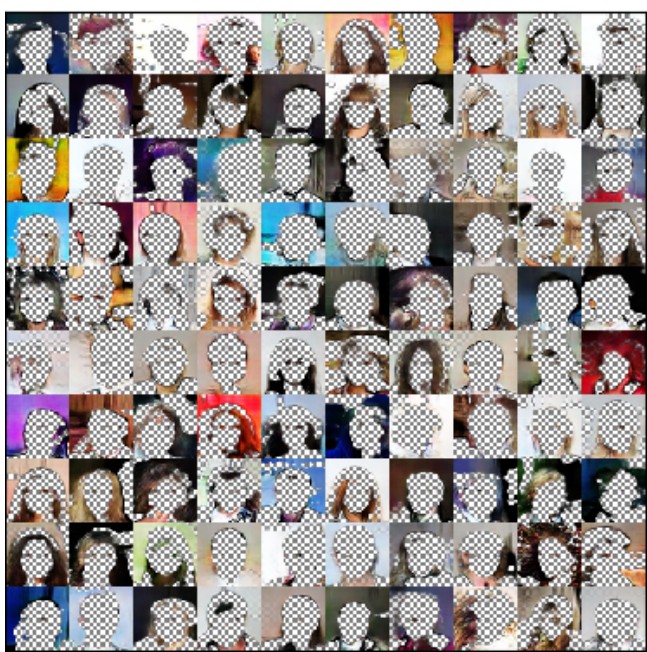

Figure 9: generated samples from $G_1$ for CelebA dataset

Trevor Darrell and Alexander Pentland. Robust estimation of a multi-layered motion representation. In *Visual Motion, 1991., Proceedings of the IEEE Workshop on*, pp. 173–178. IEEE, 1991.

Emily L Denton, Soumith Chintala, Rob Fergus, et al. Deep generative image models using alaplacian pyramid of adversarial networks. In *Advances in neural information processing systems*, pp. 1486–1494, 2015.

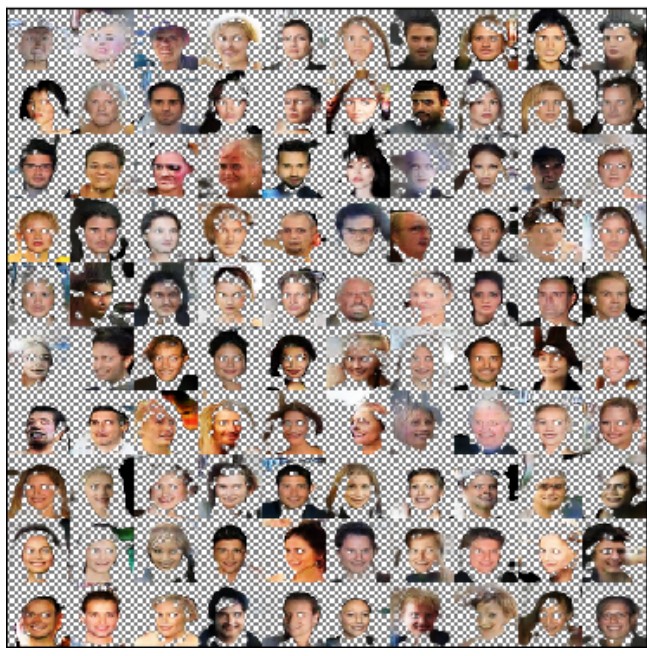

Figure 10: generated sample from $G_2$ for CelebA dataset

Ian Goodfellow. Nips 2016 tutorial: Generative adversarial networks. *arXiv preprint arXiv:1701.00160*, 2016.

Ian Goodfellow, Jean Pouget-Abadie, Mehdi Mirza, Bing Xu, David Warde-Farley, Sherjil Ozair, Aaron Courville, and Yoshua Bengio. Generative adversarial nets. In *Advances in neural information processing systems*, pp. 2672–2680, 2014.

Ishaan Gulrajani, Faruk Ahmed, Martin Arjovsky, Vincent Dumoulin, and Aaron Courville. Improved training of wasserstein gans. *arXiv preprint arXiv:1704.00028*, 2017.

Daniel Jiwoong Im, Chris Dongjoo Kim, Hui Jiang, and Roland Memisevic. Generating images with recurrent adversarial networks. *arXiv preprint arXiv:1602.05110*, 2016.

Phillip Isola and Ce Liu. Scene collaging: Analysis and synthesis of natural images with semantic layers. In *Proceedings of the IEEE International Conference on Computer Vision*, pp. 3048–3055, 2013.

Nebojsa Jojic and Brendan J Frey. Learning flexible sprites in video layers. In *Computer Vision and Pattern Recognition, 2001. CVPR 2001. Proceedings of the 2001 IEEE Computer Society Conference on*, volume 1, pp. I–I. IEEE, 2001.

Takuhiro Kaneko, Kaoru Hiramatsu, and Kunio Kashino. Generative attribute controller with conditional filtered generative adversarial networks. In *Proceedings of the IEEE Conference on Computer Vision and Pattern Recognition*, pp. 6089–6098, 2017.

Anitha Kannan, Nebojsa Jojic, and Brendan J Frey. Generative model for layers of appearance and deformation. In *Aistats*, volume 2005, pp. 1, 2005.

Hanock Kwak and Byoung-Tak Zhang. Generating images part by part with composite generative adversarial networks. *arXiv preprint arXiv:1607.05387*, 2016.

Yann LeCun, Léon Bottou, Yoshua Bengio, and Patrick Haffner. Gradient-based learning applied to document recognition. *Proceedings of the IEEE*, 86(11):2278–2324, 1998.

Ziwei Liu, Ping Luo, Xiaogang Wang, and Xiaoou Tang. Deep learning face attributes in the wild. In *Proceedings of International Conference on Computer Vision (ICCV)*, 2015.

Elman Mansimov, Emilio Parisotto, Jimmy Lei Ba, and Ruslan Salakhutdinov. Generating images from captions with attention. *arXiv preprint arXiv:1511.02793*, 2015.

Yuval Netzer, Tao Wang, Adam Coates, Alessandro Bissacco, Bo Wu, and Andrew Y Ng. Reading digits in natural images with unsupervised feature learning. In *NIPS workshop on deep learning and unsupervised feature learning*, volume 2011, pp. 5, 2011.

Augustus Odena, Christopher Olah, and Jonathon Shlens. Conditional image synthesis with auxiliary classifier gans. *arXiv preprint arXiv:1610.09585*, 2016.

Alec Radford, Luke Metz, and Soumith Chintala. Unsupervised representation learning with deep convolutional generative adversarial networks. *arXiv preprint arXiv:1511.06434*, 2015.

Scott Reed, Zeynep Akata, Xinchen Yan, Lajanugen Logeswaran, Bernt Schiele, and Honglak Lee. Generative adversarial text to image synthesis. *arXiv preprint arXiv:1605.05396*, 2016a.

Scott E Reed, Zeynep Akata, Santosh Mohan, Samuel Tenka, Bernt Schiele, and Honglak Lee. Learning what and where to draw. In *Advances in Neural Information Processing Systems*, pp. 217–225, 2016b.

Carl Vondrick, Hamed Pirsiavash, and Antonio Torralba. Generating videos with scene dynamics. In *Advances In Neural Information Processing Systems*, pp. 613–621, 2016.

John YA Wang and Edward H Adelson. Representing moving images with layers. *IEEE Transactions on Image Processing*, 3(5):625–638, 1994.

Xiaolong Wang and Abhinav Gupta. Generative image modeling using style and structure adversarial networks. In *European Conference on Computer Vision*, pp. 318–335. Springer, 2016.

Xinchen Yan, Jimei Yang, Kihyuk Sohn, and Honglak Lee. Attribute2image: Conditional image generation from visual attributes. In *European Conference on Computer Vision*, pp. 776–791. Springer, 2016.

Jianwei Yang, Anitha Kannan, Dhruv Batra, and Devi Parikh. Lr-gan: Layered recursive generative adversarial networks for image generation. *arXiv preprint arXiv:1703.01560*, 2017.

Junbo Zhao, Michael Mathieu, and Yann LeCun. Energy-based generative adversarial network. *arXiv preprint arXiv:1609.03126*, 2016.

Jun-Yan Zhu, Philipp Krähenbühl, Eli Shechtman, and Alexei A Efros. Generative visual manipulation on the natural image manifold. In *European Conference on Computer Vision*, pp. 597–613. Springer, 2016.

