# OpenReview forum: "3C-GAN: AN CONDITION-CONTEXT-COMPOSITE GENERATIVE ADVERSARIAL NETWORKS FOR GENERATING IMAGES SEPARATELY"
_ICLR.cc/2018/Conference — Reject_

### Official Review · AnonReviewer3 · 2017-11-14
**Review of 3C-GAN**

**Rating:** 5
**Confidence:** 5

**Review:**

Summary: This paper studied the conditional image generation with two-stream generative adversarial networks. More specifically, this paper proposed an unsupervised learning approach to generate (1) foreground region conditioned on class label and (2) background region without semantic meaning in the label. During training, two generators are competing against each other to hallucinate foreground region and background region with a physical gating operation. An auxiliary “label difference cost” was further introduced to encourage class information captured by the foreground generator. Experiments on MNIST, SVHN, and CelebA datasets demonstrated promising generation results with the unsupervised two-stream generation pipeline.

== Novelty/Significance ==
Controllable image generation is an important task in representation learning and computer vision. I also like the unsupervised learning through gating function and label difference cost. However, considering many other related work mentioned by the paper, the novelty in this paper is quite limited. For example, layered generation (Section 2.2.1) has been explored in Yan et al 2016 (VAEs) and Vondrick et al 2016 (GANs).

== Detailed comments ==
The proposed two-stream model is developed with the following two assumptions: (1) Single object in the scene; and (2) Class information is provided for the foreground/object region. Although the proposed method learns to distinguish foreground and background in an unsupervised fashion, it is limited in terms of applicability and generalizability. For example, I am not convinced if the two-stream generation pipeline can work well on more challenging datasets such as MS-COCO, LSUN, and ImageNet.

Given the proposed method is controllable image generation, I would assume to see the following ablation studies: keeping two latent variables from (z_u, z_l, z_v) fixed, while gradually changing the value of the other latent variable. However, I didn’t see such detailed analysis as in the other papers on controllable image generation.

In Figure 7 and Figure 10, the boundary between foreground and background region is not very sharp. It looks like equation (5) and (6)  are insufficient for foreground and background separation (triplet/margin loss could work better). Also, in CelebA experiment, it is not a well defined experimental setting since only binary label (smiling/non-smiling) is conditioned. Is it possible to use all the binary attributes in the dataset.

Also, please either provide more qualitative examples or provide some type of quantitative evaluations (through user study , dataset statistics, or down-stream recognition tasks).

Overall, I believe the paper is interesting but not ready for publication. I encourage authors to investigate (1) more generic layered generation process and (2) better unsupervised boundary separation. Hopefully, the suggested studies will improve the quality of the paper in the future submission.

== Presentation ==
The paper is readable but not well polished.

-- In Figure 1, the “G1” on the right should be “G2”;
-- Section 2.2.1, “X_f” should be “x_f”;
-- the motivation of having “z_v” should be introduced earlier;
-- Section 2.2.4, please use either “alpha” or “\alpha” but not both;
-- Section 3.3, the dataset information is incorrect: “20599 images” should be “202599 images”;

Missing reference:
-- Neural Face Editing with Intrinsic Image Disentangling, Shu et al. In CVPR 2017.
-- Domain Separation Networks, Bousmalis et al. In NIPS 2016.
-- Unsupervised Image-to-Image Translation Networks, Liu et al. In NIPS 2017.

---

### Official Review · AnonReviewer1 · 2017-11-26
**Review from AnonReviewer1**

**Rating:** 4
**Confidence:** 4

**Review:**

[Overview]

This paper proposed a new generative adversarial network, called 3C-GAN for generating images in a composite manner. In 3C-GAN, the authors exploited two generators, one (G1) is for generating context images, and the other one (G2) is for generating semantic contents. To generate the semantic contents, the authors introduced a conditional GAN scheme, to force the generated images to match the annotations. After generating both parts in parallel, they are combined using alpha blending to compose the final image. This generated image is then sent to the discriminator. The experiments were conducted on three datasets, MNIST, SVHN and MS-CelebA. The authors showed qualitative results on all three datasets, demonstrating that AC-GAN could disentangle the context part from the semantic part in an image, and generate them separately.

[Strenghts]

This paper introduced a layered-wise image generation, which decomposed the image into two separate parts: context part, and semantic part. Corresponding to these two parts are two generators. To ensure this, the authors introduced three strategies:

1. Adding semantic labels: the authors used image semantic labels as the input and then exploited a conditional GAN to enforce one of the generators to generate semantic parts of images. As usual, the label information was added as the input of generator and discriminator as well.

2. Adding label difference cost: the intuition behind this loss is that changing the label condition should merely affect the output of G2. Based on this, outputs of Gc should not change much when flipping the input labels.

3. Adding exclusive prior: the prior is that the masks of context part (m1) and semantic part (m2) should be exclusive to each other. Therefore, the authors added another loss to reduce the sum of component-wise multiplication between m1 and m2.

Decomposing the semantic part from the context part in an image based on a generative model is an interesting problem. However, to my opinion, completing it without any supervision is challenging and meaningless. In this paper, the authors proposed a conditional way to generate images compositionally. It is an interesting extension of previous works, such as Kwak & Zhang (2016) and Yang (2017).

[Weaknesses]

This paper proposed an interesting and intuitive image generation model. However, there are several weaknesses existed:

1. There is no quantitative evaluation and comparisons. From the limited qualitative results shown in Fig.2-10, we can hardly get a comprehensive sense about the model performance. The authors should present some quantitative evaluations in the paper, which are more persuasive than a number of examples. To do that, I suggest the authors exploited evaluation metrics, such as Inception Score to evaluate the overall generation performance. Also, in Yang (2017) the authors proposed adversarial divergence, which is suitable for evaluating the conditional generation. Hence, I suggest the authors use a similar way to evaluate the classification performance of classification model trained on the generated images. This should be a good indicator to show whether the proposed 3C-GAN could generate more realistic images which facilitate the training of a classifier.

2. The authors should try more complicated datasets, like CIFAR-10. Recently, CIFAR-10 has become a popular dataset as a testbed for evaluating various GANs. It is easy to train since its low resolution, but also means a lot since it a relative complicated scene. I would suggest the authors also run the experiments on CIFAR-10.

3. The authors did not perform any ablation study. Apart from several generation results based on 3C-GAN, iIcould not found any generation results from ablated models. As such, I can hardly get a sense of the effects of different losses and know about the relative performance in the whole GAN spectrum. I strongly suggest the authors add some ablation studies. The authors should at least compare with one-layer conditional GAN.

4. The proposed model merely showed two-layer generation results. There might be two reasons: one is that it is hard to extend it to more layer generation as I know, and the other one reason is the inflexible formulation to compose an image in 2.2.1 and formula (6). The authors should try some datasets like MNIST-TWO in Yang (2017) for demonstration.

5. Please show f1, m1, f2, m2 separately, instead of showing the blending results in Fig3, Fig4, Fig6, Fig7, Fig9, and Fig10. I would like to see what kind of context image and foreground image 3C-GAN has generated so that I can compare it with previous works like Kwak & Zhang (2016) and Yang (2017).

6. I did not understand very well the label difference loss in (5). Reducing the different between G_c(z_u, z_v, z_l) and G_c(z_u, z_v, z_l^f) seems not be able to force G1 and G2 to generate different parts of an image. G2 takes all the duty  can still obtain a lower L_ld. From my point of view, the loss should be added to G1 to make G1 less prone to the variation of label information.

7. Minor typos and textual errors. In Fig.1, should the right generator be G2 rather than G1? In 2.1.3 and 2.2.1, please add numbers to the equations.

[Summary]

This paper proposed an interesting way of generating images, called 3C-GAN. It generates images in a layer-wise manner. To separate the context and semantic part in an image, the authors introduced several new techniques to enforce the generators in the model undertake different duties. In the experiments, the authors showed qualitative results on three datasets, MNIST, SVHN and CelebA. However, as I pointed out above, the paper missed quantitative evaluation and comparison, and ablation study. Taking all these into account, I think this paper still needs more works to make it solid and comprehensive before being accepted.

---

### Official Review · AnonReviewer2 · 2017-11-27
**Due to poor experimental validation, the merit of the proposed method is unknown.**

**Rating:** 4
**Confidence:** 5

**Review:**


- Paper summary

The paper proposes a label-conditional GAN generator architecture and a GAN training objective for the image modeling task. The proposed GAN generator consists of two components where one focuses on generating foreground while the other focuses on generating background. The GAN training objective function utilizing 3 conditional classifier. It is shown that through combining the generator architecture and the GAN training objective function, one can learn a foreground--background decomposed generative model in an unsupervised manner. The paper shows results on the MNIST, SVHN, and Celebrity Faces datasets.

- Poor experimental validation

While it is interesting to know that a foreground--background decomposed generative model can be learned in an unsupervised manner, it is clear how this capability can help practical applications, especially no such examples are shown in the paper. The paper also fails to provide any quantitative evaluation of the proposed method. For example, the paper will be more interesting if inception scores were shown for various challenging datasets.  In additional, there is no ablation study analyzing impacts of each design choices. As a result, the paper carries very little scientific value.

---

### Decision · Program_Chairs · 2018-01-29
**ICLR 2018 Conference Acceptance Decision**

**Decision:**

Reject

**Comment:**

The paper presents a layered image generation model  (e.g., foreground vs background) using GANs. The high-level idea is interesting, but novelty is somewhat limited. For example, layered generation with VAE/GAN has been explored in Yan et al. 2016 (VAEs) and Vondrick et al. 2016 (GANs). In addition, there are earlier works for unsupervised learning of foreground/background generative models (e.g., Le Roux et al., Sohn et al.). Another critical problem is that only qualitative results on relatively simple datasets (e.g., MNIST, SVHN, CelebA) are provided as experimental results. More quantitative evaluations and additional experiments on more challenging datasets will strengthen the paper.

* N. Le Roux, N. Heess, J. Shotton, J. Winn; Learning a generative model of images by factoring appearance and shape; Neural Computation 23(3): 593-650, 2011.
** Sohn, K., Zhou, G., Lee, C., & Lee, H. Learning and selecting features jointly with point-wise gated Boltzmann machines. ICML 2013.